# Design and Experiment of a Lifting Tool for Hoisting Offshore Single-Pile Foundations

**Bo Zhang** [1,*] **, Hexuan Chen** [1] **, Tao Wang** [2] **and Zhuo Wang** [1]

1   College of Mechanical and Electrical Engineering, Harbin Engineering University, Harbin 150001, China; chenyin@hrbeu.edu.cn (H.C.); wangzhuo_heu@hrbeu.edu.cn (Z.W.)
2   School of Mechanical Engineering, Hebei University of Technology, Tianjin 300130, China; 18846166436@hrbeu.edu.cn
*   Correspondence: zhangbo_heu@hrbeu.edu.cn

**Abstract:** Experiments with a cam-type clamp tool were carried out to overcome the difficulty of transporting and installing large-diameter mono-piles for offshore wind turbines. Using the experiments method to design a small wedge-type clamping mechanism and using cam teeth made of 40Cr material resulted in the maximum friction for the mechanism. A single clamping design was created for the cam-type clamp tool to hoist mono-piles for offshore wind turbines. Through force analysis and Automatic Dynamics Analysis of Mechanical System (ADAMS) dynamic simulation of the lifting tool, it was calculated that the clamping force of the lifting tool meets application requirements. A prototype was built in order to carry out an experiment in which the lifting tool hoisted a mono-pile. It was concluded from the experiment that the proposed design of the lifting tool is feasible in practical applications.

**Keywords:** offshore wind turbine; cam-type clamp; structure design; force analysis; ADAMS simulation; prototype testing

## 1. Introduction

In recent years, wind power has become the fastest developing new energy technology; however, it cannot be applied widely on land due to the fact that there is little space for wind farms and to the great noise produced when generating electricity [1]. Wind turbines consist of a frame to orientate the blades, a pile gripper, a subsea structure for installing a pile, and a monopole [2]. The structure of the rotor blades and that of the engine depend on the installed capacity of the wind turbine. At present, turbine structures can be categorized as mono-pile, gravity foundation, suction foundation, multi-pile foundation, floating foundation, etc. [3]. The simple structure of the mono-pile enables it to be applied widely in engineering, but in consideration of the fact that the pile installation will have lateral force vibration and the seabed requires high stability [4], mono-piles are not available in sea areas with a water depth of more than 25 m [5]. Further, the surface disk and deep anchor problems cause degenerate conditions under appropriate limit conditions [6]. By pouring concrete caissons into the seabed, the gravity foundation method makes the whole turbine remain vertical using the gravity of the turbine, but subsea operations indicate that the overall cost is high, and this method is limited to applications in shallow water [7]. In the theory of the suction method, the negative pressure produced by pumping water out of the underwater steel caisson is employed to make components such as the engine adsorb to the seabed [8]. There is also a single-pile installation technology which can transfer energy with a higher frequency of striking, which reduces energy consumption and noise [9]. The vertical compliance of the pile foundation shows great dependence on the embedded depth, excitation frequency, permeability, and soil stratification [10].

Multi-pile foundations utilize three to four steel piles to build a supporting structure combined with concrete. The technology is still at the experimental stage and not yet

used in engineering projects [11]. All the methods mentioned above are only applicable to coastal waters. Nowadays, turbines can be built on floating platforms [12]. Similar to those of an offshore oil drilling platform, the structures of the floating platform include the pressure sensor type, tension leg type and floating box type [13].

Compared with other types of supporting foundation for turbines, the turbine foundation with a mono-pile has been widely used because of its simple structure and low cost [14]. Under the same cyclic load, the cyclic resistance capacity of a large-diameter single pile with greater stiffness is better than that of a conventional long pile [15]. The asymmetric problem of rocking rotation of a circular rigid disk embedded in a finite depth of a transversely isotropic half-space was analytically addressed. The jump behavior in the results at the edge of the rigid disk for the case of an infinitesimal embedment was highlighted analytically [16]. At present, the maximum diameter of a mono-pile is 5 m, and the pile foundation diameter of 6~8 m has great development prospects [17]. Due to the fact that the mono-pile will have a huge impact during the installation, the pile foundation is often designed as a smooth hollow cylinder in order to avoid the phenomenon of stress concentration [18]. Due to the specific structure of the pile, a general lifting tool cannot be directly utilized, so it is necessary to design a special lifting tool to hoist the mono-pile. At present, up-ending unilateral or bilateral clamping tools and lifting tools for piles produced by the Dutch company IHC-MERWEDE can reliably complete hoisting operations of 250 and 500 tons for mono-pile foundations [19].

At present, lifting tools for piles which are suitable for engineering applications are relatively rare. In view of this situation, in this paper, we studied the lifting tools for mono-pile turbines and developed a novel lifting tool, which functions by clamping via cam, to solve this engineering problem.

## 2. Experimental Methods for the Clamping Mechanism

The research method in this paper was based on carrying out a basic experiment to determine the teeth of the cam, then carrying out three-dimensional design and motion simulation analysis, and finally processing the prototype to simulate a lifting experiment. The coefficient of friction between the contact surfaces of the two materials varies with the type of material. For the lifting tool, the coefficient of friction between the clamping device and the pile foundation should be as large as possible. The material of the pile foundation is generally stainless steel. The clamping force of the tool can be increased by utilizing a tool of the appropriate material to increase the coefficient of friction on the clamping surface [20]. The structure of the small wedge clamping mechanism is shown in Figure 1.

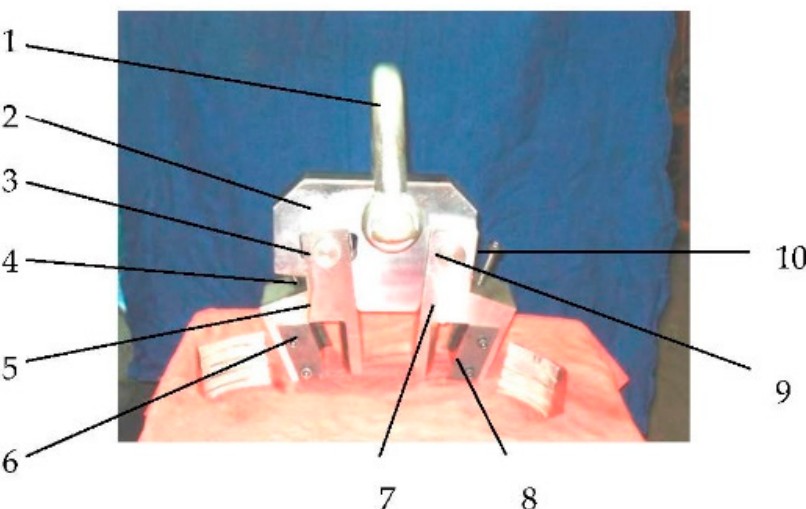

**Figure 1.** Structure of the wedge clamping mechanism.

The mechanism consists of lugs (1), fixed diagonal slides (2), left swivel pins (3), side nail bolts (4), side moving wedge blocks (5), side compression inserts (6), a right inner moving wedge (7), a right pressing fast (8), a right swivel pin (9), a right nail bolt (10), and other components. The end of the pile is installed in the gap between the side moving wedge blocks (5), the right inner moving wedge (7), block 6, and block 8. The method of installation of the clamping mechanism is shown in Figure 2. The structural details of the wedge clamping mechanism are shown in Figure 3.

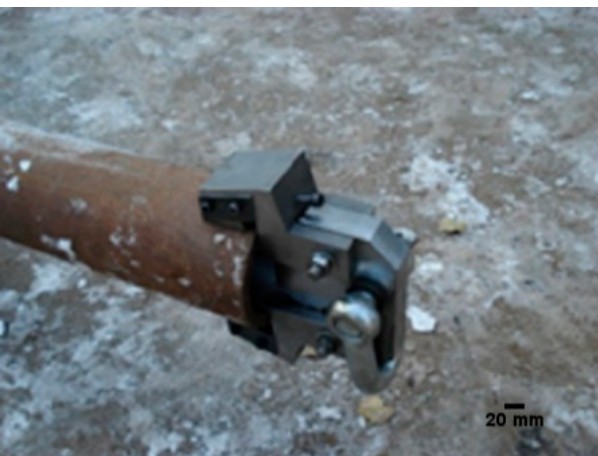

**Figure 2.** The installation method of the wedge clamping mechanism.

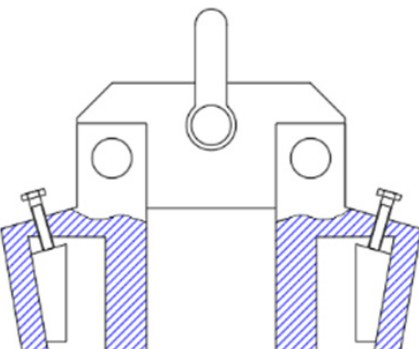

**Figure 3.** The structural details of the wedge clamping mechanism.

The main parts of the clamping mechanism are clamping block 6 and clamping block 8. These blocks are in contact with the piles and clamp them. The blocks were designed to be replaceable, so the clamping effect produced by blocks of different materials could be studied by changing the materials of the blocks. In addition, the impact of the tooth shape of the cam used to hoist the pile could also be studied by testing different shapes in the clamping experiment.

Clamping blocks made of 40Cr, T8, and 9SiCr were each utilized in pile hoisting tests. The tooth angle of the upper teeth of the wedge block was 60°, and five teeth were evenly distributed on each wedge block. A picture of the experiment in progress is shown in Figure 4.

The experiment demonstrated that when combined with the clamping mechanism, all the blocks made of those three materials mentioned above could be utilized to clamp and hoist the steel piles. Through the indentation of the clamping tool on the outer surface of the pile, it was observed that the three kinds of clamping block led to local deformation of the pile. The tooth tip was embedded in the base material of the foundation pile, but the whole pile was not deformed, which indicates that the embedding of the tooth increases the friction, so that the loading force of the mechanism was improved [21].

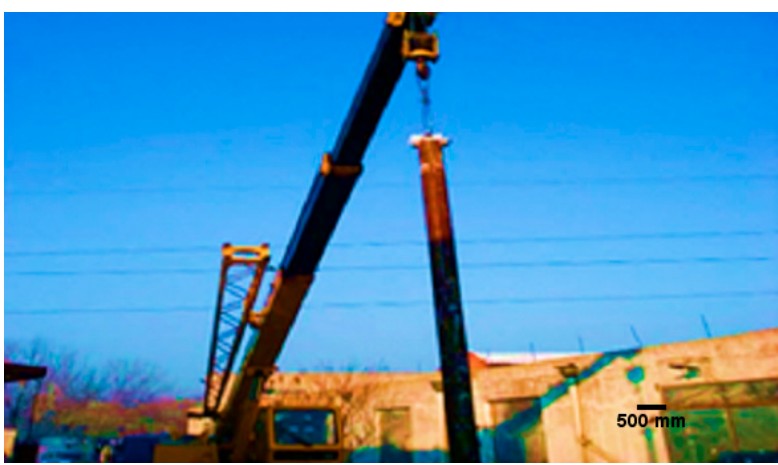

**Figure 4.** A picture of the hoisting experiment in progress.

In order to further explore the effect of the cam tooth material on the clamping force, both an experiment on the loading force for the clamping mechanism and an experiment on the ultimate bearing capacity of the three kinds of materials were carried out. The maximum tension was recorded when the wedge was unfastened by the clamping force. The relationship between the clamping force and the maximum tension is expressed in Table 1.

**Table 1.** Results obtained for wedge mechanisms with different parameters.

| Number | The Shape and Material of the Teeth | Clamping Force (KN) | The Max Value of Tension (KN) | Equivalent Coefficient of Friction |
|---|---|---|---|---|
| 1 | 9SiCr (Serrated shape) | 78.4 | 76.9 | 0.98 |
| 2 | 40Cr (Blade shape) | 156.8 | 184.2 | 1.18 |
| 3 | T8 (Blade shape) | 156.8 | 139.2 | 0.89 |

The results of the bearing experiment are also expressed in Figure 5.

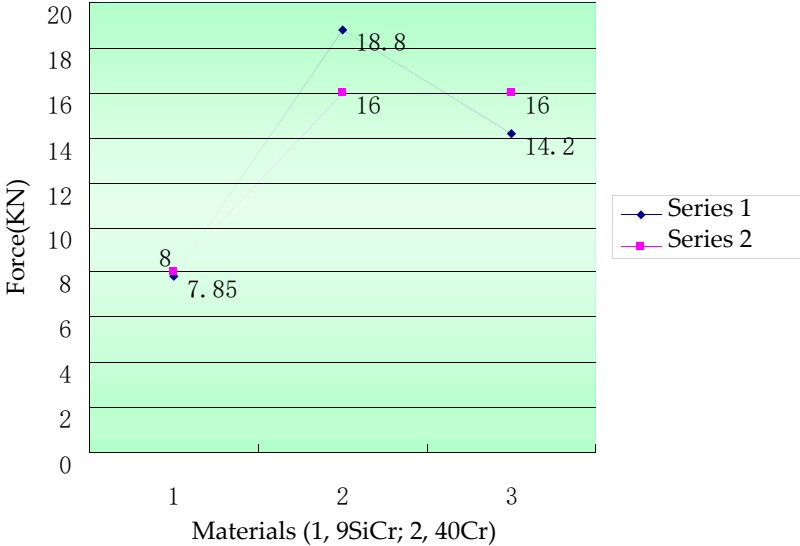

**Figure 5.** Line chart of the relationship between the clamping force, tensile force, and the material of the clamping tooth.

In Figure 5, the blue curve shows the relationship between the tensile force and the material of the clamping tooth (series 1), while the purple curve shows the relationship between the clamping force and the material of the clamping tooth (series 2). Due to the difference in shapes and materials of the teeth, different clamping forces were applied during the experiment. The maximum tension was observed when the clamping tool was disengaged. After that, the coefficient of friction was calculated and the dimensionless coefficient was obtained, which was independent of the force.

It can be seen from Table 1 and Figure 5 that the maximum tensile force is related to the material and shape of the clamping tooth when the clamping force is equal. The clamping tooth made of 40Cr provided the largest coefficient of friction, which means that the 40Cr material has a greater clamping capacity than 9SiCr and T8. The coefficient of friction provided by 40Cr was 1.18, which indicates that the relationship between the pressure and the friction is not static friction. The teeth of the cam were embedded in the inner part of the pile, which produced local plastic deformation, increasing the coefficient of friction.

Based on the experiment and analysis mentioned above, under the same load, 40Cr material can provide greater friction and clamping force. For the following cam clamp design, we thus used 40Cr as the material.

## 3. Design and Simulation of the Cam-Type Clamp Tool

The designed lifting tool is intended to be used to hoist the mono-piles of offshore wind turbines. The lifting tool is suitable for piles with weight of 100~300 t, diameter of 4000~5000 mm, and thickness within the range of 20~70 mm. The clamping part of the lifting tool was designed as a cam. The clamping method is a combination of hydraulic drive and the use of the weight of the pile.

The three-dimensional structure of the mechanism is shown in Figure 6.

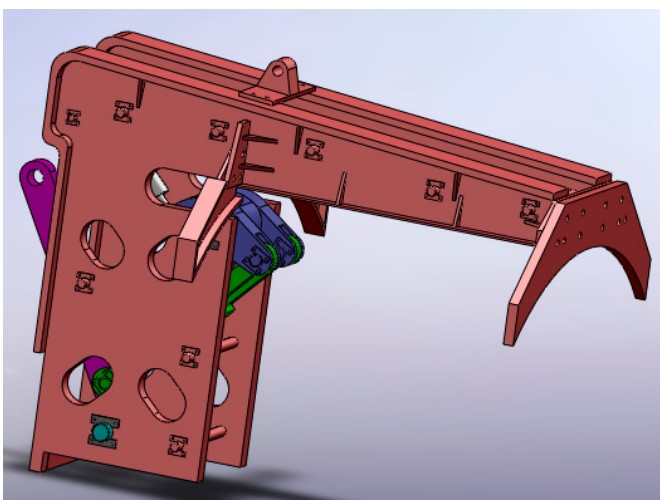

**Figure 6.** The three-dimensional structure of the lifting tool.

The internal structure of the lifting tool is shown in Figure 7. It consists of the main lifting lug (part No.1), hydraulic cylinder base-connecting shaft (part No.2), hydraulic cylinder (part No.3), main frame (part No.4), frame-connecting shaft (part No.5), auxiliary lifting lug (part No.6), hydraulic cylinder clamping cam (part No.7), self-weight clamping cam (part No.8), front-end outer clamping plate (part No.10), rear-middle outer clamping plate (part No.11), rear-middle outer clamping plate connecting bolt (part No.12), main lifting lug limit shaft (part No.13), main lifting lug and main lifting lug and self-weight cam connecting rod-connecting bolt (part No.14), main frame-connecting shaft (part No.15), the initial positioning spring (part No.16) and the dead weight cam-connecting rod (part No.17). Part No.9 is the steel pipe.

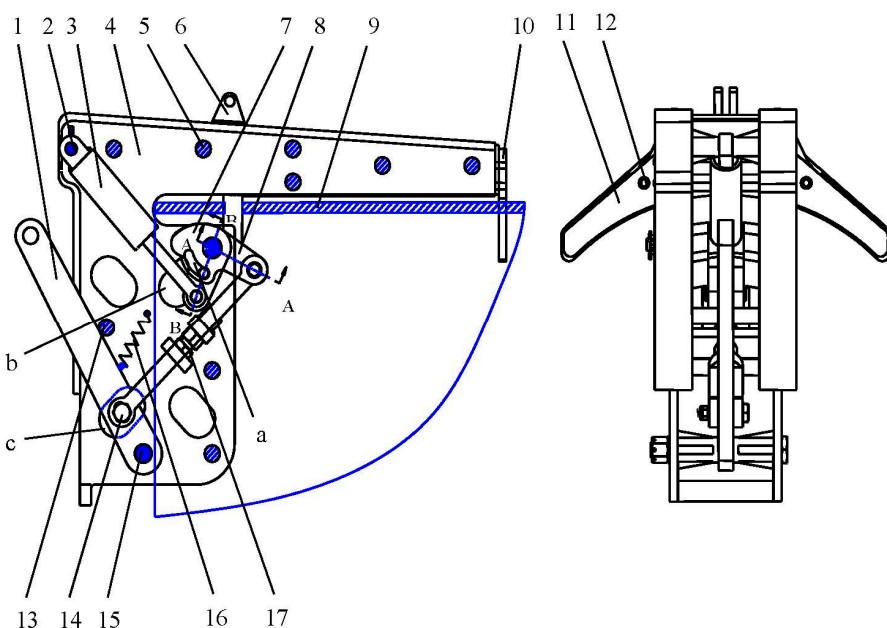

**Figure 7.** The internal structure of the lifting tool.

The working process of the lifting tool is to use the hydraulic cylinder to drive cam part No.9 to complete the pre-clamping of the pile wall in order to fix the lifting tool. Then, tension is applied at the main lug to lift up the pile gradually. In this process, the self-weight driving cam, part No.10, gradually puts force on the inner wall of the pile. When the pile is vertical to the ground, the lifting tool can provide enough clamping force to lift up the whole pile. After the completion of the lifting operation, the lifting tool is removed by unloading the hydraulic cylinder.

The whole process of lifting is shown in Figure 8a–d. The operation process is divided into four steps: placing the lifting tool, lifting the pile foundation, fully hoisting the monopile, and releasing the lifting tool.

The operation of placing the lifting tool is shown in Figure 8a. First, the hydraulic cylinder is unloaded and the lifting tool placed at the end part of the pile. The detailed steps are as follows. The piston rod of the hydraulic cylinder (part No.3) is extended, driving the clamping cam (part No.7) of the hydraulic cylinder and the self-weight driving cam (part No.8) to rotate counterclockwise to the limit position, where the distance between the clamping cam part No.7 and part No.8 and the inner wall of the pile is the maximum value, with the clamping tool loosened. Then, the lifting tool is placed at the end of pile (part No.9) by the crane applying tensile force at the secondary lug (part No.7). The control valve of the hydraulic system will convert the direction, and the rod of the hydraulic cylinder (part No.3) will move back, driving the clamping cam (part No.8) of the hydraulic cylinder to rotate clockwise to clamp the inner wall of the pile (part No.9). Thus, the first process of clamping the steel pile by the lifting tool is finished, as shown in Figure 8b. Pin part No.17 and cam part No.7 of the hydraulic cylinder cannot drive the self-weight cam, because it has slid to the limit position of cam part No.8.

When the pile is completely vertical to the ground, as shown in Figure 8c, the self-weight driving cam (part No.8) will rotate to the limit position, and the clamping force reaches the maximum value. The tensile force applied on the axis of the main lug (part No.1) is also applied on the axis of the pile. At this moment, the pile is lifted up completely, and the engineer can move the pile to the working position. When the pile is at the expected position, the lifting tool should be removed, as shown in Figure 8d. The lifting tool can be unloaded by controlling the hydraulic cylinder to retract the rod. In this process, the rod will drive the hydraulic cylinder clamping cam to rotate counterclockwise around the pin. At the same time, the limit position pin of the cam, part No.17, will also drive the self-weight driving cam, part No.8, to rotate in the same direction. As a result, the inner

wall of the pile is released by the cams. The main lug (part No.1) will rotate to the initial position. Then, tensile force is applied at the main lug. The initial position of the main lug is also the center of gravity of the lifting tool, so the lifting tool can be removed easily.

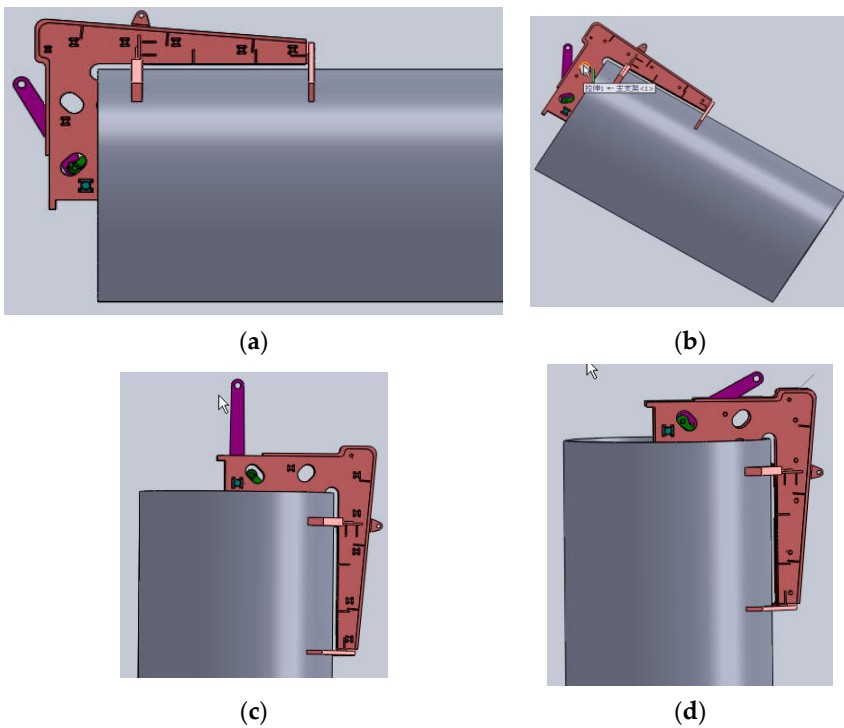

(**a**)　　　　　　　　　　　　　　　　　　　　　　(**b**)

(**c**)　　　　　　　　　　　　　　　　　　　　　　(**d**)

**Figure 8.** The process of the lifting operation. (**a**) Unload the hydraulic cylinder and place the lifting tool (**b**) Lift the pipe. (**c**) Hang the steel pipe (**d**) Remove the lifting tool.

The structure of the lifting tool was designed to increase the clamping force. The length of the link and the position of the pin were calculated to satisfy the law of leverage. During the process of the lifting operation, the clamping force provided by the lifting tool to the inner wall of the pile is almost 6 times the gravity of the pile, which ensures that the lifting tool is reliable in practical engineering applications. The force of the clamping part of the lifting tool is shown in Figure 9.

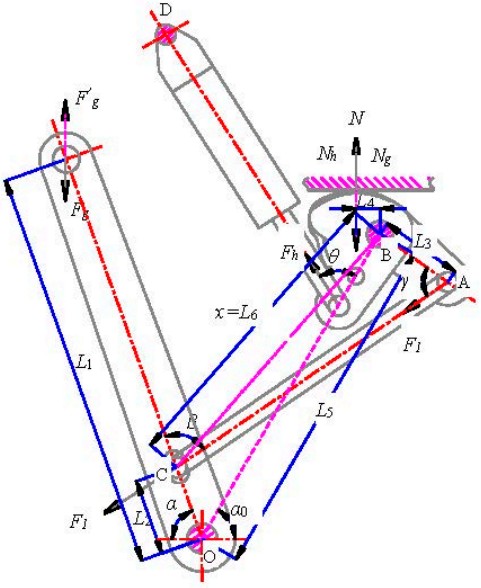

**Figure 9.** Force analysis of the clamping mechanism.

The tensile force provided by the crane is denoted $F_g'$. The direction of the applied force $F_g'$ is vertical upward. The clamping force, denoted as $N$, is a combination of the clamping force provided by the hydraulic cylinder, $N_h$, and that provided by the self-weight driving cam, $N_g$.

As shown in Figure 9, the degree of freedom of the clamping mechanism is 1. The angle between the main lug and the horizontal line is $\alpha$. The angle between the main lug and the stretch-out and draw-back pole is $\beta$.

The angle between the piston rod of the hydraulic cylinder and the axis of the cam is $\theta$. The angle between the telescopic pole and the axis of the self-weight driving cam is $\gamma$. The arms of force are $L_1$, $L_2$, $L_3$, and $L_4$. The equilibrium of the mechanism can be expressed in Equation (1).

$$\begin{aligned} F_g \cdot \cos\alpha \cdot L_1 &= F_1 \cdot \sin\beta \cdot L_2 \\ F_1 \cdot \sin\gamma \cdot L_3 &= N_g \cdot L_4 \\ F_h \cdot \sin\theta \cdot L_3 &= N_h \cdot L_4 \end{aligned} \tag{1}$$

The clamping force can be expressed as Equation (2), based on Equation (1).

$$\begin{aligned} N_h &= \frac{F_h \cdot \sin\theta \cdot L_3}{L_4} \\ N_g &= \frac{F_g \cdot \cos\alpha \cdot \sin\gamma \cdot L_1 \cdot L_3}{\sin\beta \cdot L_2 \cdot L_4} \\ N &= N_h + N_g \end{aligned} \tag{2}$$

The force produced by the piston of the hydraulic cylinder is denoted as $F_h$. It was found that the force of the hydraulic cylinder $N_h$ relates to the angle $\theta$ and the arms ratio $L_3/L_4$, and the force $N_g$ provided by the self-weight driving cam relates to the angles $\alpha$, $\gamma$, $\beta$ and the arms ratios $L_3/L_4$, $L_1/L_2$. From Equation (2), the relationship between $N_g$ and $\alpha$ can be expressed as Equation (3)

$$N_g = \frac{F_g L_1 L_3 \cos\alpha \sqrt{4L_3^2 L_{AC}^2 + x^2 - L_3^2 - L_{AC}^2}}{2L_2 L_4 L_{AC} \sin(\pi - \arccos(\frac{x^2 + L_{AC}^2 - L_3^2}{2xL_{AC}}) + \arccos(\frac{x^2 + L_2^2 - L_5^2}{2xL_2}))} \tag{3}$$

where $x$ is equal to $L_6$, and it can be calculated to be $\sqrt{L_2^2 + L_5^2 - 2L_2 L_5 \cos(\pi - \alpha_0 - \alpha)}$.

At the beginning of the clamping operation, the geometric parameters of the lifting tool can be expressed as follows: $\alpha = 70°$, $\beta = 75°$, $\gamma = 70°$, $\theta = 63°$, $L_1 = 610$ mm, $L_2 = 115$ mm, $L_3 = 125$ mm, $L_4 = 35$ mm. By substituting all these parameters into Equation (3), $N_h$ and $N_g$ can be calculated. After the calculation, the relationships of $N_h$ with $F_h$ and $N_g$ with $F_g$ can be expressed: $N_h = 3.18F_h$, $N_g = 6.3F_g$. This shows that the clamping tool meets the requirements concerning frictional force, and the hydraulic cylinder helps to increase the clamping force. The parameters of the mechanism change when the whole pile is completely lifted up. These new parameters are as follows: $\alpha = 0°$, $\beta = 130°$, $\gamma = 130°$, $\theta = 118°$. After the recalculation, the result was $N_h = 3.15F_h$, $N_g = 18.9F_g$, with the clamping force produced by self-weight becoming three times the initial clamping force, which further ensures the safety of the operation.

In order to further study the reliability of the lifting tool, a dynamic simulation of the lifting process was carried out in the ADAMS environment. The result of the simulation is shown in Figure 10a–d.

In Figure 10a, the cam of the hydraulic cylinder clamps but does not lift the pile. In Figure 10b, with the pile lifted and clamped by the cam of the hydraulic cylinder, the clamping force provided by the self-weight driving cam increases with increasing lifted angle of the pile. In Figure 10c, when the pile is vertical to the ground, the clamping force of the self-weight driving cam reaches the maximum value. In Figure 10d, the pile is completely lifted up. In the simulation, the lifting tool reached the expectation of hoisting the pile, which shows that the design of the lifting tool was successful.

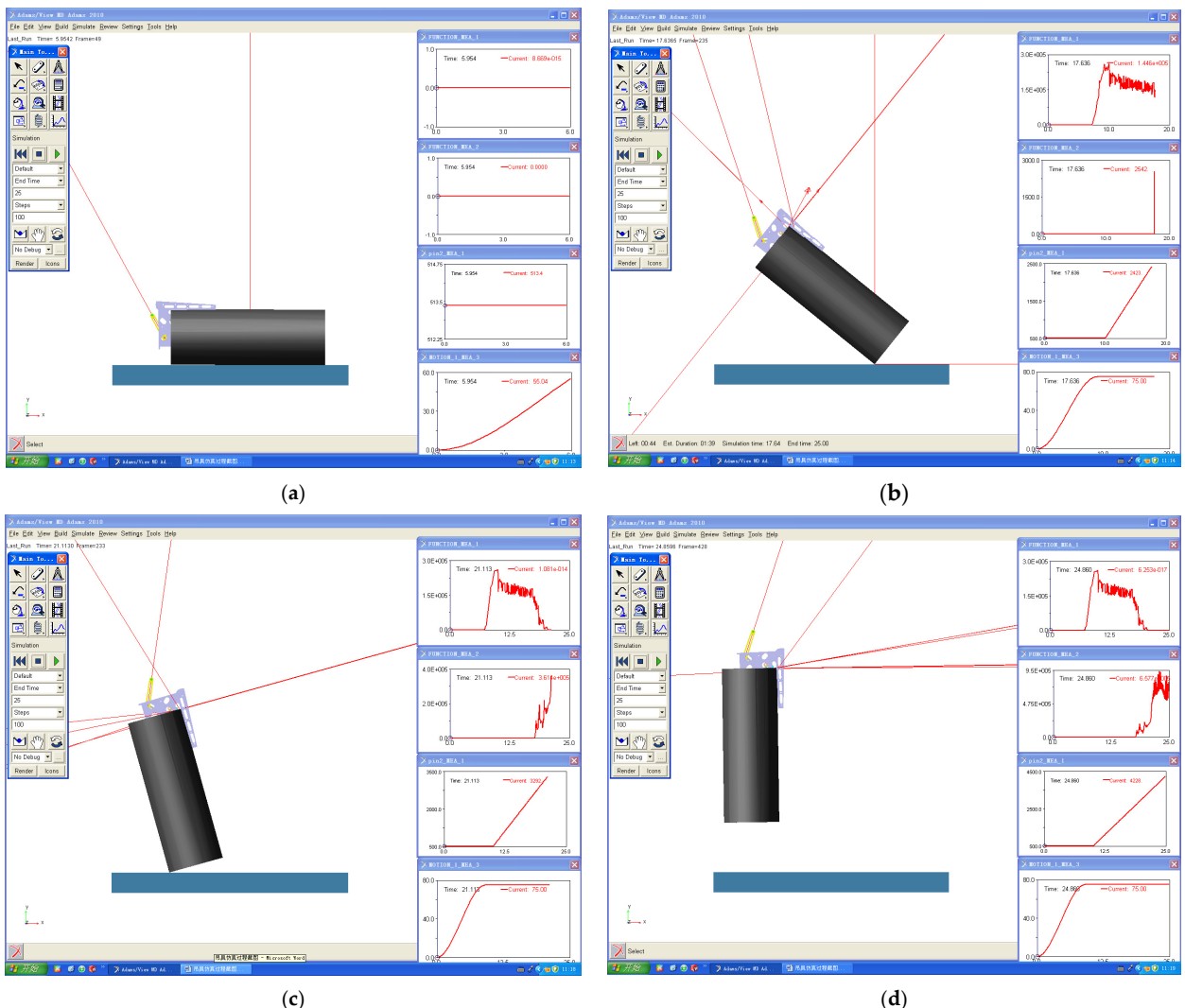

**Figure 10.** A dynamic simulation of the lifting process. (**a**) Clamping steel pipe with spreader (**b**) Lifting the steel pipe (**c**) State before lifting the steel pipe vertically (**d**) After lifting the steel pipe.

The software used can measure the force between the clamping cam of the hydraulic cylinder and the inner wall of the pile and that between the self-weight driving cam and the pile. The result is shown in Figure 11.

In Figure 11, the red curve represents the change, over time, in the force between the clamping cam of the hydraulic cylinder and the inner wall of the pile. At the seventh second, the clamping cam of the hydraulic cylinder came into contact with the surface of the inner wall of the pile; thus, a clamping force appeared. With the movement of the piston, the clamping force between the cam of hydraulic cylinder and the inner wall of the pile increased and reached its maximum of $2.89 \times 10^5$ N at the tenth second. Compared to the gravity of the pile, the clamping force was 2.89 times bigger than the gravity, which matches with the analysis introduced before. In the whole lifting process, the clamping force was in the range of $[1.33 \times 10^5 \, N, \, 2.20 \times 10^5 \, N]$. The blue curve shows the trend of the force between the self-weight driving cam and the pile. In this level, the self-weight driving cam began to work, and the total clamping force started to increase again. The maximum value of clamping force provided by the self-weight driving cam was $9.21 \times 10^5$ N.

The total clamping force consists of the clamping force from the clamping cam of the hydraulic cylinder and the force from the self-weight driving cam. The change in the total force over time is shown in Figure 12.

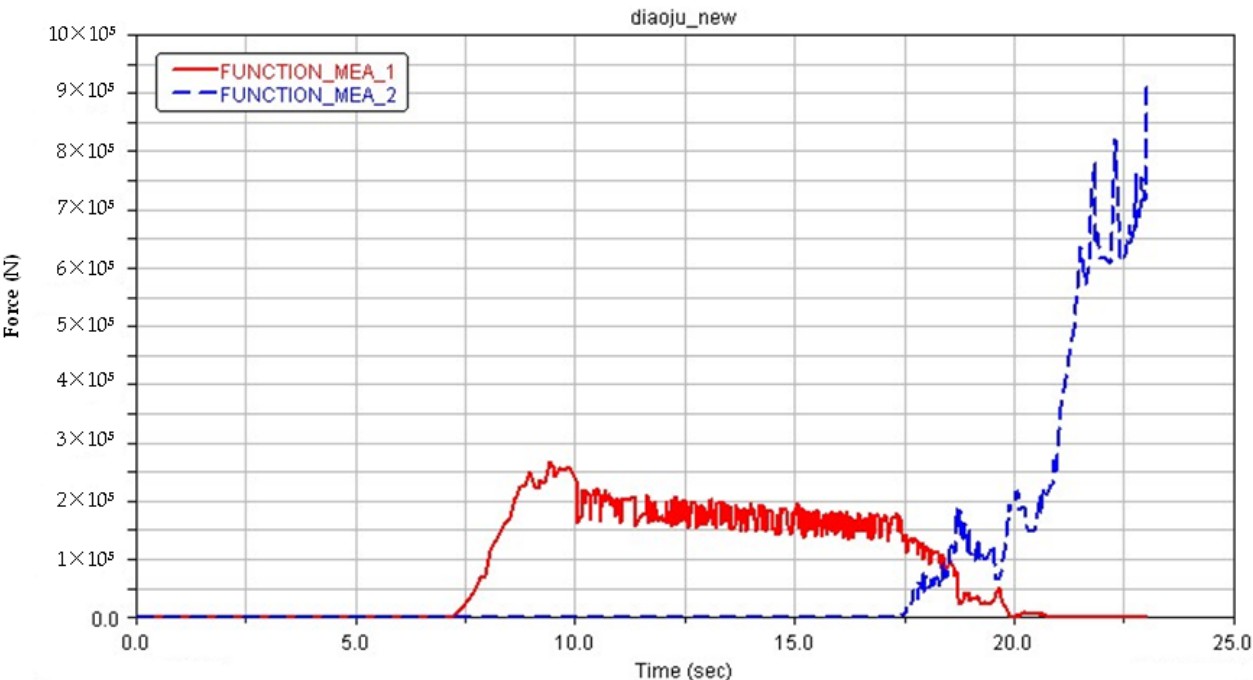

**Figure 11.** The trends of the force between the hydraulic cylinder clamping cam and the inner wall of the pile and the force between the self-weight driving cam and the pile.

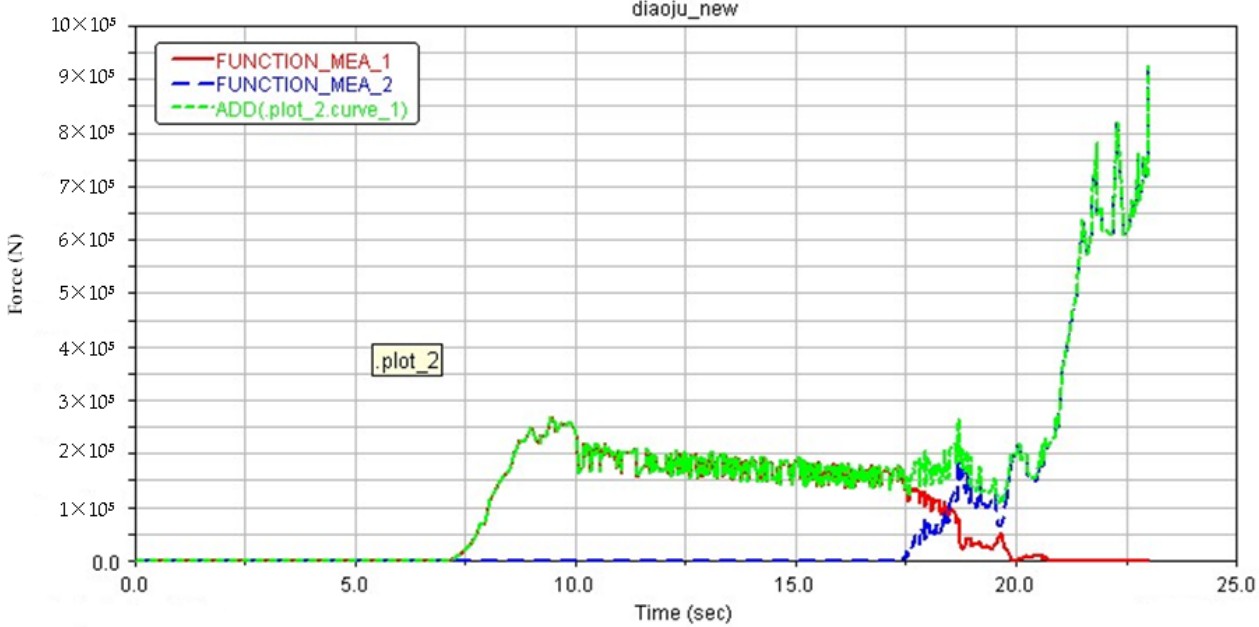

**Figure 12.** The changing of the total clamping force with time.

In Figure 12, the green curve represents the total clamping force. In the time range of [0 s, 7 s], the total clamping force was 0 N, which means that neither of the cams worked. In the time range of [7 s, 10 s], the clamping cam of the hydraulic cylinder began to work, and the clamping force increased to $2.89 \times 10^5$ N. In the time range of [10 s, 20 s], the clamping force was in the range of [$1.33 \times 10^5$ N, $2.20 \times 10^5$ N], which means that the clamping cam of the hydraulic cylinder was working, while the self-weight driving cam did not work. After 20 s, the clamping force continued to increase, which shows that the self-weight driving cam was clamping the pile. When the clamping force reached the maximum value, the lifting tool had lifted up the pile completely.

Based on the analysis and simulation mentioned above, the designed lifting tool can reliably fulfill the task and meet the engineering requirements.

## 4. Prototype Experiment of the Clamp Tool

A prototype machine of the lifting tool was manufactured to test the reliability of the lifting tool. The machine is shown in Figure 13. The prototype machine was manufactured in a 1:5 reduction ratio to the lifting tool applied in the engineering projection. The diameter of the pile foundation in the experiment was 1219.2 mm (48 inches), the thickness of the inner wall of the pile was in the range of 20~25 mm, and the weight of the pile was 5 t. The material of the pile was Q235b; the process of the experiment is shown in Figure 14a–d.

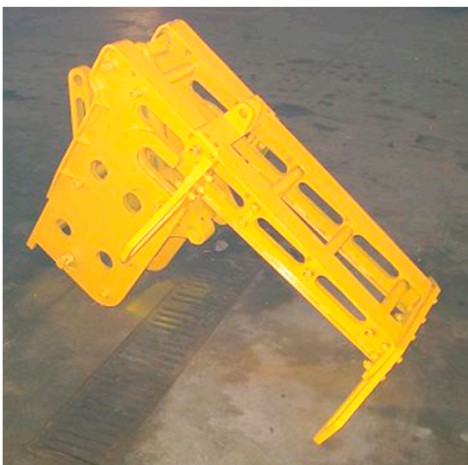

**Figure 13.** The prototype of the lifting tool.

The experiment consisted of placing the lifting tool, lifting the pile, fully hoisting the pile, and removing the lifting tool. The lifting tool was installed at the end part of the pile by the crane applying tensile force on the secondary lug. Then, the rod of the clamping hydraulic cylinder pulled the cam to clamp the inner wall of the pile, as shown in Figure 14a.

The working position of the crane to the main lug was changed and tensile force was applied to lift up the pile, as shown in Figure 14b. The gravity of the pile pulled the self-weight driving cam to rotate in order to clamp the pile through the transmission mechanism. Thus, the force provided by the clamping cam of the hydraulic cylinder and the self-weight driving cam lifted up the pile.

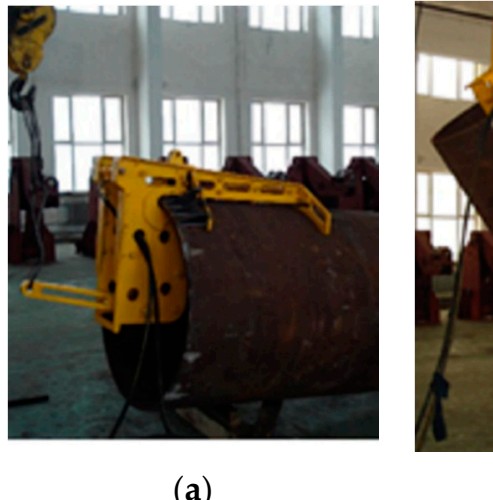 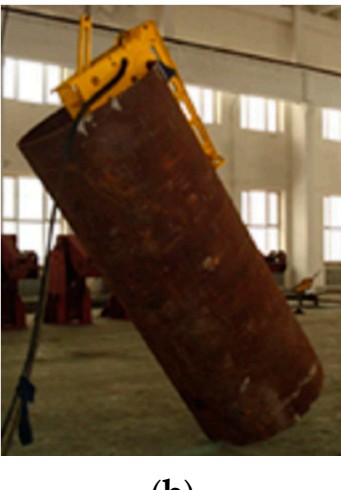

(**a**)                                          (**b**)

**Figure 14.** *Cont.*

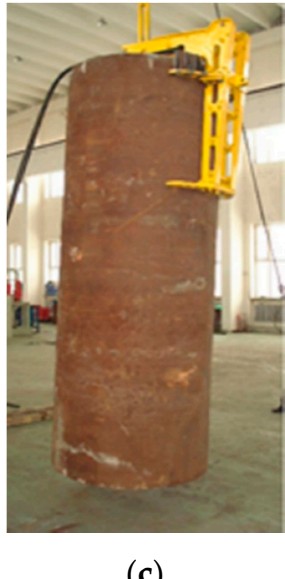
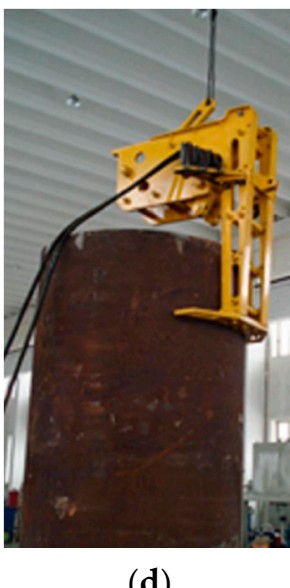

**(c)**            **(d)**

**Figure 14.** The process of the prototype machine. (**a**) Unload the hydraulic cylinder and place the lifting tool (**b**) Lift the pipe. (**c**) Hang the steel pipe (**d**) Remove the lifting tool.

When the axis of the pile was vertical to the ground, as shown in Figure 14c, the clamping force provided by the self-weight driving cam reached the maximum value, and the clamping teeth of the lifting tool were embedded in the pile to increase the clamping force. As a result, the pile foundation was completely hoisted above the ground.

When the pile was moved to the expected position, the rod of the clamping hydraulic cylinder was moved back. At the same time, both the clamping cam of the hydraulic cylinder and the self-weight driving cam rotated to the initial position. The lifting tool did not come into contact with the inner wall of the lifting tool. Tensile force was applied at the main lug, and the lifting tool could be removed easily, as shown in Figure 14d.

During the experiment, the lifting tool reliably finished the hoisting task. Strong, flexible, and convenient, the lifting tool has great value in engineering applications.

## 5. Conclusions

In this paper, a cam-type lifting tool was designed to complete the operation of carrying the mono-piles of turbines. The lifting tool makes use of a hydraulic cylinder and the gravity of the mono-pile to hoist the pile foundation. The design was verified to be feasible through static analysis and dynamic simulation via ADAMS. After an experiment with a prototype machine, the design was proved to be reliable in practical engineering applications. With this lifting tool, the pile foundation can be lifted, moved, put in place, and automatically unlocked without interruption. Compared with cable hoisting, it can shorten the construction period, thereby reducing the required number of workers. In addition, this lifting tool can reduce the time taken to install pile foundations, advancing the grouting operation and reducing the whole construction time, so construction costs can be reduced.

**Author Contributions:** The first author, B.Z., conceived the framework of the article and wrote the article; the second author, H.C., analyzed the experimental method on the clamping mechanism of spreaders; the third author, T.W., studied on the design and simulation of the cam type clamp tool; Z.W. carried out the prototype experiment of the clamp tool. All authors have read and agreed to the published version of the manuscript.

**Funding:** This paper was funded by NSFC (Contract name: Research on ultimate bearing capacity and parametric design for the grouted clamps strengthening the partial damaged structure of jacket pipes, Contract number: 51879063, and Contract name: Research of Analysis and Experiments of Gripping and Bearing Mechanism for Large-scale Holding and Lifting Tools on Ocean Foundation Piles, Contract number: 51479043). The views expressed here were the authors' alone.

**Institutional Review Board Statement:** The study was conducted according to the guidelines of the NSFC and approved by Harbin Engineering University (protocol code 51879063 and date 1 January 2019).

**Informed Consent Statement:** Informed consent was obtained from all subjects involved in the study.

**Data Availability Statement:** We do not need to upload data. Readers can contact authors if they have questions. The authors can explain the data analysis.

**Conflicts of Interest:** The authors declare no conflict of interest.

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
