# Peer review of "Design and Experiment of a Lifting Tool for Hoisting Offshore Single-Pile Foundations"

_machines, doi:10.3390/machines9020029_

Round 1

Reviewer 1 Report

This manuscript studied the lifting fixture for offshore single piles, and designed a small wedge clamping mechanism and a cam clamping mechanism. The clamping force was analyzed by Adams, and the feasibility of the clamping tool was verified through experiments. The paper is practical, rigorous in analysis and innovative. Minor points: 1. The description and working principle of the small wedge clamping structure described in Figure 1 are not detailed enough. The design details and clamping effect of the wedge clamping mechanism cannot be clearly obtained simply from the introduction in Figure 1 and Figure 2. The exploded diagram of structural details and the schematic diagram of the clamping process must be added to enhance the reader's understanding! 2. In Figure 3 and Figure 4, there is a sample error in the analysis of a single experimental result. It is not statistically significant. It is recommended to perform statistical analysis for multiple experiments to obtain more rigorous and credible experimental results! 3. According to the experimental results of Fig. 3 and Fig. 4, the analysis results of the friction force of the cam teeth embedded in the pile are put forward. The introduction of specific tooth structure should be added in the article to increase the reader's understanding. 4. The second and third sections of the thesis are not sufficiently coherent. It is necessary to add instructions on the design and research of two different sizes and structures of fixtures. Highlight its internal connections. 5. According to the force analysis in Fig. 8, the Fg direction in Fig. 9 should be vertical. Please explain the reason for the change in the direction of Fg in the simulation result in Fig. 9? 6. The images and graphs in the text are not sufficiently clear. It is recommended to use vector graphics to improve the quality of the paper.

Author Response

I am very grateful to your comments for the manuscript. According you’re your advice, we amended the relevant part in manuscript. Some of your questions were answered below.

To Reviewer #1:

Specific issues

  • Opinion 1: The description and working principle of the small wedge clamping structure described in Figure 1 are not detailed enough. The design details and clamping effect of the wedge clamping mechanism cannot be clearly obtained simply from the introduction in Figure 1 and Figure 2. The exploded diagram of structural details and the schematic diagram of the clamping process must be added to enhance the reader's understanding!

Responses to 1: As shown in lines 78-79 and Figure 3, we added a schematic diagram of the wedge block to facilitate the reader's understanding.

  • Opinion 2: In Figure 3 and Figure 4, there is a sample error in the analysis of a single experimental result. It is not statistically significant. It is recommended to perform statistical analysis for multiple experiments to obtain more rigorous and credible experimental results!

Responses to 2: Fig. 3 and Fig. 4 are experiments on the lifting mechanism of wedge clamping mechanism. The clamping blocks made of 40Cr, T8 and 9SiCr were utilized respectively in test about hoisting the piles. The tooth Angle of the upper teeth of the wedge block is 60°, and five teeth are evenly distributed on each wedge block. The clamping tooth made of 40Cr can provide the largest coefficient of friction, which means that the material of 40Cr has a higher clamping capacity than the material of 9SiCr and T8. The coefficient of friction provided by 40Cr is 1.18, which indicates that the relationship between the pressure and the friction is not a static friction. The teeth of cam have been embedded in the inner of the pile, which produced the local plastic deformation, increasing the coefficient of friction.

  • Opinion 3: According to the experimental results of Fig. 3 and Fig. 4, the analysis results of the friction force of the cam teeth embedded in the pile are put forward. The introduction of specific tooth structure should be added in the article to increase the reader's understanding.

Responses to 3: As shown in lines 92-93, we added a description of the tooth shape and the number of teeth on the wedge block to facilitate the reader's understanding.

  • Opinion 4: The second and third sections of the thesis are not sufficiently coherent. It is necessary to add instructions on the design and research of two different sizes and structures of fixtures. Highlight its internal connections.

Responses to 4: As shown in lines 126-128, the foregoing wedge-shaped block clamping experiment determined that 40Cr was the material to provide the maximum clamping force, and 40Cr was used as the material for the cam-type clamping method designed below.

  • Opinion 5: According to the force analysis in Fig. 8, the Fg direction in Fig. 9 should be vertical. Please explain the reason for the change in the direction of Fg in the simulation result in Fig. 9?

Responses to 5: The figure shows the vertical condition of the pile foundation. In the model, the main lifting lug is not parallel to the axis of the pile foundation, and the pile foundation should have a certain inclination Angle, so the force line of the main lifting lug is not vertical.

  • Opinion 6: The images and graphs in the text are not sufficiently clear. It is recommended to use vector graphics to improve the quality of the paper.

Responses to 6: Let's change to high-definition image columns, as shown in Figure 6, Figure 7. Figure 8, Figure 9, Figure 11 ,Figure 12 and  Figure 13.

************************************************

We would like to express our great appreciation to you and reviewers for comments on our paper. Looking forward to hearing from you.

Thank you and best regards.

Yours sincerely,

Bo Zhang

Bo Zhang

Eail: zhangbo_heu@hrbeu.edu.cn

Jan.12.2021

Reviewer 2 Report

Dear Authors

I think that the work is good, well presented and well written, so I think that it could be published.

Other considerations are about the lack of novelty and scientific interest, nowadays.

So, I suggest to enrich contents and literature, maybe reading the following:

"The "C-triplex" approach to design of CFRP transport-category airplane structures" of Piancastelli et al.

Best Regards

Reviewers

Author Response

I am very grateful to your comments for the manuscript. According you’re your advice, we amended the relevant part in manuscript. Some of your questions were answered below.

To Reviewer #2:

Specific issues

Thank you for your comments on this paper and we look forward to your next comments!

Responses to: Thank you for your suggestion, composite material is a good direction, we added some content and enriched the references.

************************************************

We would like to express our great appreciation to you and reviewers for comments on our paper. Looking forward to hearing from you.

Thank you and best regards.

Yours sincerely,

Bo Zhang

Bo Zhang

Eail: zhangbo_heu@hrbeu.edu.cn

Jan.12.2021

Reviewer 3 Report

There are some points that need to be addressed before making a final decision on this manuscript.  

  1. There are some reports that analytically and numerically addressed the interaction of a foundation and its surrounding media (refer to the following references).  These works should be reviewed in the introduction of the manuscript.
    1. "Vibration analysis of a rigid circular disk embedded in a transversely isotropic solid." Journal of Engineering Mechanics 140.7 (2014): 04014048.  
    2. "Rocking rotation of a rigid disk embedded in a transversely isotropic half-space." Civil Engineering Infrastructures Journal 47.1 (2014): 125-138.
    3. "Vertical dynamic analysis of a rigid disk embedded in layered saturated soils with compressible fluid." Computers and Geotechnics 119 (2020): 103347.
    4. "Forced vertical vibration of rigid discs with arbitrary embedment." Journal of Engineering Mechanics 117.11 (1991): 2527-2548.
  2. Proper scale bars should be added to some of the figures of the manuscript - figs. 1-3
  3. The English of the manuscript should be checked again.  There are some grammatical errors in the manuscript. 
  4. The quality of the figures is not really good as it is hard for the reviewer to read some of the figures. 

Author Response

I am very grateful to your comments for the manuscript. According you’re your advice, we amended the relevant part in manuscript. Some of your questions were answered below.

To Reviewer #3:

Specific issues

Opinion 1: There are some reports that analytically and numerically addressed the interaction of a foundation and its surrounding media (refer to the following references).  These works should be reviewed in the introduction of the manuscript.

    1. "Vibration analysis of a rigid circular disk embedded in a transversely isotropic solid." Journal of Engineering Mechanics 140.7 (2014): 04014048.  
    2. "Rocking rotation of a rigid disk embedded in a transversely isotropic half-space." Civil Engineering Infrastructures Journal 47.1 (2014): 125-138.
    3. "Vertical dynamic analysis of a rigid disk embedded in layered saturated soils with compressible fluid." Computers and Geotechnics 119 (2020): 103347.
    4. "Forced vertical vibration of rigid discs with arbitrary embedment." Journal of Engineering Mechanics 117.11 (1991): 2527-2548.

Responses to 1: As shown in lines 32-36, 43-45,according to the suggestions, we added the above articles to the introduction to enrich the references.

  • Opinion 2: Proper scale bars should be added to some of the figures of the manuscript - figs. 1-3

Responses to 2: As shown in the article, we added the scale bars to the above figure.

  • Opinion 3: The English of the manuscript should be checked again. There are some grammatical errors in the manuscript.

Responses to 3: Improved English language expression as suggested. If there is no content problem in the revised article, we will entrust the editor of this journal to edit the grammar and sentence pattern of the article.

  • Opinion 4: The quality of the figures is not really good as it is hard for the reviewer to read some of the figures.

Responses to 4: : Let's change to high-definition image columns, as shown in Figure 6, Figure 7. Figure 8, Figure 9, Figure 11 ,Figure 12 and  Figure 13.

************************************************

We would like to express our great appreciation to you and reviewers for comments on our paper. Looking forward to hearing from you.

Thank you and best regards.

Yours sincerely,

Bo Zhang

Bo Zhang

Eail: zhangbo_heu@hrbeu.edu.cn

Jan.12.2021

Round 2

Reviewer 3 Report

Authors have not fully incorporated my comments to the manuscript; just a few papers suggested in my first comments were reviewed in the introduction.  The reviewer doesn’t have extra time to review the literature for authors.  Therefore, all the suggested references should be highlighted in the introduction of tue manuscript.  All comments should be fully addressed before making a decision on this manuscript. 

Author Response

I am very grateful to your comments for the manuscript. According you’re your advice, we amended the relevant part in manuscript. Some of your questions were answered below.

To Reviewer #3:

Opinion 1of The First review: There are some reports that analytically and numerically addressed the interaction of a foundation and its surrounding media (refer to the following references).  These works should be reviewed in the introduction of the manuscript.

    1. "Vibration analysis of a rigid circular disk embedded in a transversely isotropic solid." Journal of Engineering Mechanics 140.7 (2014): 04014048.  
    2. "Rocking rotation of a rigid disk embedded in a transversely isotropic half-space." Civil Engineering Infrastructures Journal 47.1 (2014): 125-138.
    3. "Vertical dynamic analysis of a rigid disk embedded in layered saturated soils with compressible fluid." Computers and Geotechnics 119 (2020): 103347.
    4. "Forced vertical vibration of rigid discs with arbitrary embedment." Journal of Engineering Mechanics 117.11 (1991): 2527-2548.

Opinion of The Second review:

Authors have not fully incorporated my comments to the manuscript; just a few papers suggested in my first comments were reviewed in the introduction.  The reviewer doesn’t have extra time to review the literature for authors.  Therefore, all the suggested references should be highlighted in the introduction of tue manuscript.  All comments should be fully addressed before making a decision on this manuscript. 

Responses to: According to the first comment and recommended references, we revised the article again. Four references were cited in the paper, these articles are No.4, No.6, No.10 and No.16, please see the red font. Please review again.

In addition, we have also revised the article according to the format of the journal, please also review.

************************************************

We would like to express our great appreciation to you and reviewers for comments on our paper. Looking forward to hearing from you.

Thank you and best regards.

Yours sincerely,

Bo Zhang

Bo Zhang

Eail: zhangbo_heu@hrbeu.edu.cn

Jan.20.2021
